# Emerging Insights into the Role of the Microbiome in Brain Gliomas: A Systematic Review of Recent Evidence

**DOI:** 10.3390/ijms27010444

**Published:** 2025-12-31

**Authors:** Piotr Dubiński, Martyna Odzimek-Rajska, Sebastian Podlewski, Waldemar Brola

**Affiliations:** 1Doctoral School, The Jan Kochanowski University, Żeromskiego 5, 25-369 Kielce, Poland; piotr.dubinski@gmail.com (P.D.); odzimek.martyna@onet.pl (M.O.-R.); 2Clinical Department of Neurosurgery and Spine Surgery, Provincial Hospital in Kielce, Grunwaldzka 45, 25-736 Kielce, Poland; sebastian.podlewski@wp.pl; 3Institute of Health Sciences, Collegium Medicum, The Jan Kochanowski University, Al. IX Wieków Kielc 19A, 25-516 Kielce, Poland

**Keywords:** microbiota, microbiome, dysbiosis, glioma, glioblastoma

## Abstract

Gliomas, particularly glioblastoma multiforme, remain among the most lethal brain tumours despite multimodal therapy. Increasing evidence indicates that systemic factors, including the gut microbiota, may influence glioma progression through immune, metabolic, and neurochemical pathways. We conducted a comprehensive systematic review in accordance with the Preferred Reporting Items for Systematic Reviews and Meta-Analyses (PRISMA) 2020 guidelines to synthesize recent evidence on the role of gut and intratumoral microbiota in glioma biology. Peer-reviewed studies published within the last five years were identified through structured searches of major biomedical databases, and original studies using human cohorts, animal models, or Mendelian randomization approaches were included. The 17 studies met the eligibility criteria. Glioma was consistently associated with gut dysbiosis characterized by a reduced *Firmicutes*:*Bacteroidetes* ratio and enrichment of *Verrucomicrobia*, particularly *Akkermansia*, alongside decreased short-chain fatty acids and altered neurotransmitter profiles, contributing to neuroinflammation, immune suppression, and blood–brain barrier dysfunction. Antigenic mimicry by Bacteroidetes-derived peptides may impair antitumour T-cell responses, while intratumoral *Fusobacteriota* and *Proteobacteria* appear to promote angiogenesis and pro-inflammatory chemokine expression. In contrast, SCFA-producing taxa such as *Ruminococcaceae* and probiotic genera including *Lactobacillus* and *Bifidobacterium* show protective associations. Evidence is limited by small cohorts and methodological heterogeneity. Standardized humanized models and integrated multi-omics approaches are required to clarify causal mechanisms and support microbiome-targeted therapies in glioma.

## 1. Introduction

Gliomas are primary tumours of the central nervous system, originating from glial cells, which, according to World Health Organization (WHO) data, constitute 30% of all primary brain tumours in adults. Among brain tumours of glial origin, glioblastoma multiforme (GBM) represents the most malignant and lethal subtype, characterized by rapid progression, poor therapeutic response, and a median overall survival of less than 20 months, with fewer than 5% of patients surviving five years, despite multimodal treatment approaches involving neurosurgical intervention, radiotherapy and chemotherapy with temozolomide (TMZ) [1]. In the WHO classification of gliomas, based primarily on molecular patterns, as well as genetic and histopathological features, there are four grades indicating different degrees of malignancy, and GBM is classified as grade IV [1]. While significant strides have been made in the molecular characterization of gliomas, including the identification of IDH mutations and epigenetic signatures, the understanding of glioma pathogenesis remains incomplete, particularly regarding environmental and systemic contributors. The cause of brain gliomas has not been definitively determined. Possible risk factors include high radiation doses, human cytomegalovirus (CMV), obesity, and a family history of cancer [1]. A growing body of research underscores the role of the gut microbiota in shaping neurological health and disease. The microbiome represents a collection of microorganisms existing both externally and internally within the human body, referred to as microbiota, together with their surrounding environmental conditions and the entire combined genetic pool, forming a reciprocal relationship with the host and co-creating one common ecosystem. The gastrointestinal tract and, above all, the large intestine serves as the primary site for colonization by microorganisms. Recent scholarly attention in the realms of medicine and physiology has concentrated on investigating the role of microbiota in the pathogenesis of neurological disorders because of bidirectional communication with the central nervous system (CNS) via the gut–brain axis [2,3]. This interaction is known to influence neurodevelopment, neuroinflammation, and neurodegenerative processes [3,4]. The implications of such systemic crosstalk have extended into the realm of oncology, where the microbiota has emerged as a modulator of cancer progression, immune response, and treatment efficacy [4,5]. Recent investigations have revealed significant alterations in the composition and metabolic activity of the gut microbiota in patients with CNS tumours, including gliomas [6]. Dysbiosis, defined as the inability of microbiota to maintain a dynamic composition and balance, may influence gliomagenesis through multiple pathways, such as immune modulation, metabolite signalling, e.g., short-chain fatty acids (SCFAs), and epigenetic remodelling of the tumour microenvironment [4,6]. In murine models, glioma development has been associated with specific shifts in microbial taxa, reflected in the disturbance in the *Firmicutes*:*Bacteroidetes* (F/B) ratio and faecal metabolomic profiles. Thus, the notion that gut microbes may contribute to tumour immune escape is supported [7]. Furthermore, the discovery of bacterial DNA and lipopolysaccharide (LPS) within glioma tissues has prompted inquiries into the possibility of a microbiome component within the CNS itself, potentially acting in concert with gut-derived signals [7]. This raises the possibility that both local and systemic microbial influences may contribute to the complex immunological and metabolic image of gliomas. Parallel studies on microbiota–immune system interactions emphasize how microbial metabolites can regulate T cell activation, microglial function, and cytokine production, all of which are pivotal in glioma progression [4,5,6]. As the field of neuro-oncology increasingly embraces the systemic nature of glioma biology, the gut–brain axis and the microbiome are gaining recognition as critical contributors. This review aims to synthesize the latest knowledge regarding the microbiome’s involvement in glioma pathophysiology, with particular emphasis on mechanisms of immune and metabolic regulation via the gut–brain axis.

The gut–brain axis (GBA) constitutes a complex, bidirectional communication network linking the gastrointestinal tract and the central nervous system. This dynamic system integrates neural, endocrine, immune, and metabolic signalling pathways, all of which are modulated by the gut microbiota [3,5]. Increasing evidence reveals that this axis plays a crucial role in regulating homeostasis, behaviour, cognition, and neuroinflammation, with significant implications for CNS disorders including gliomas [5,7]. The GBA comprises multiple interacting components: the enteric nervous system and vagus nerve, as parts of the autonomic nervous system (ANS), neuroendocrine signalling via the hypothalamic–pituitary–adrenal (HPA) axis, and the immune system [3,5]. The above-mentioned communication is facilitated through neurotransmitters, cytokines, microbial metabolites, and hormonal signals. The gut microbiota constitutes an integral component of this system, exerting significant influence over its development and functional responsiveness [3]. Key neuroactive molecules, including short-chain fatty acids such as acetate, propionate, and butyrate, as well as neurotransmitters like γ-aminobutyric acid (GABA), serotonin, and dopamine, are produced by the microbiota and are known to modulate neuronal activity and blood–brain barrier (BBB) permeability [3,4]. For instance, SCFAs have been shown to influence microglial maturation and regulate gene expression via histone deacetylase (HDAC) activity inhibition [4,6]. Moreover, SCFAs can cross the BBB and interact with immune, neural and glial cells via G protein-coupled receptors (GPCRs) and through epigenetic modifications [4]. Tryptophan metabolism represents another key channel between gut microbes and the CNS. Tryptophan, an essential amino acid, is metabolized via complex pathways into neuroactive compounds such as kynurenine, kynurenic acid, quinolinic acid, tryptamine, indole, and indole-derivatives [6]. Importantly, gliomas exploit the kynurenine pathway to suppress anti-tumour immune responses via upregulation of enzymes such as indoleamine2,3-dioxygenase1 (IDO1) and tryptophan2,3-dioxygenase (TDO), both of which are modulated by microbial signals [6]. The immune system serves as a vital intermediary in the GBA. Approximately 70% of the body’s immune lymphoid and lymphocyte cells reside in the gut-associated lymphoid tissue (GALT), and their maturation is heavily influenced by microbial antigens and metabolites [4]. These immune cells can influence distant tissues, including the brain, through cytokine secretion and activated lymphocytes trafficking. Gliomas are well known for creating an immunosuppressive microenvironment, rich in tumour-associated macrophages (TAMs), that produce immunosuppression promoting molecules such as IL-10, TGF-β, STAT3 [1,4]. Gut microbiota may exacerbate or mitigate this immunosuppression by promoting expansion of regulatory T cells (Tregs) via SCFA signalling and regulating systemic cytokine levels such as IL-10, and TGF-β [4,6]. Moreover, the microbial regulation of myeloid-derived suppressor cells (MDSCs), whose accumulation in GBM Tregs and suppresses anti-tumour T-cell activity, has been implicated in glioma development and progression [6]. Studies in glioma-bearing mice have demonstrated that tumour growth can alter the composition and function of the gut microbiome. These alterations include changes in microbial diversity and abundance of specific taxa, as well as shifts in faecal metabolomic profiles [7]. The modulation of the microbiome through antibiotics or probiotics has been shown to influence systemic immune activation, and even response to PD-L1, which is a protein found on the surface of many cells, including cancer cells, that promotes evasion of the immune system. A notable observation is the presence of bacterial DNA within glioma tissues themselves, suggesting either a low abundance intratumoral microbiome or translocation from peripheral sources [7]. However, the functional significance of these intratumoral microbiota signatures remains unclear. The neuroendocrine component of the GBA, particularly the HPA axis, further mediates stress responses that can impact glioma progression. Stress-induced glucocorticoids modulate immune surveillance and BBB integrity, potentially enabling tumour infiltration or immune evasion [3]. The HPA axis may be regulated by the microbiota through modulation of circulating levels of adrenocorticotropic hormone (ACTH), cortisol, and corticosterone via vagal afferent pathways [3,6]. Given the multifactorial involvement of the gut–brain axis in glioma pathogenesis, modulation of the microbiome is considered a promising novel therapeutic strategy. Analysis of microbial metabolites and microbiota profiling may provide valuable diagnostic and prognostic biomarkers. Restoration of microbial diversity, through prebiotics and probiotics or faecal microbiota transplantation, holds therapeutic potential in glioma treatment when implemented alongside current standard therapies [7].

## 2. Methods

### 2.1. Protocol, Reporting Framework, Research Question, and Eligibility Criteria

A systematic literature search for publications regarding microbiome and brain gliomas was conducted, spanning the period from 2020 to 2025, with the last search conducted on 31 March 2025. The search was conducted in medical literature, analysis, and retrieval system on-line (MEDLINE) and the Cochrane Central Register of Controlled Trial (Cochrane) databases, Additional sources included trial registries, reference lists of included studies, and other literature, ensuring comprehensive coverage of relevant evidence. The following search phrases were used: MEDLINE (via PubMed)—(‘microbiota’ [MeSH] or ‘microbiome’) and (‘glioma’ [MeSH] or ‘glioblastoma’ [MeSH]) yielding 56 records; and Cochrane—(‘microbiome’ or ‘microbiota’ and ‘gliomas’), yielding 0 records. The inclusion criteria comprised high-impact reviews, fundamental in vitro and in vivo experimental studies, including animal models research, statistical analyses based on Mendelian Randomization, and clinical cohort studies. The research covers phenomena related to dysbiosis in the course of gliomas, primarily microbial taxonomy, metabolic pathways, metabolomic alterations, and immunomodulation. The exclusion criteria constituted articles describing patient populations other than those with gliomas and reports that mainly dealt with aspects related to the surgical technique. Studies were independently screened by two reviewers, with disagreements resolved through discussion or a third reviewer. The materials were grouped according to study type—including in vitro experiments, in vivo animal models, human cohort studies, and Mendelian randomization analyses and by primary outcomes, such as microbial composition, metabolite profiles, and immune response markers, to allow a structured synthesis of the evidence. Extracted data were standardized, coded, and cross-verified prior to synthesis to ensure consistency across studies. Extracted data were standardized, coded, and cross-verified prior to synthesis to ensure consistency across studies. Due to the heterogeneity of outcomes and methods in the included observational studies, this systematic review employs a narrative synthesis in accordance with PRISMA 2020. Although not a scoping review, a PRISMA-ScR–inspired approach was applied to ensure transparency of scope and eligibility criteria. Heterogeneity among studies was assessed qualitatively based on differences in study design, populations, interventions, and outcomes; quantitative measures such as I^2^ were not applicable due to the narrative synthesis approach. No formal subgroup or sensitivity analyses were conducted due to the limited number of studies and heterogeneity of reported outcomes. The search strategy followed PRISMA 2020 guidelines and is illustrated in Figure 1.

### 2.2. Risk of Bias and Quality Appraisal

The methodological quality of the included studies was critically assessed according to study design. In vitro and in vivo studies were evaluated for experimental rigor, reproducibility, and the adequacy of control conditions, while observational human studies were appraised for potential sources of bias, including selection bias, measurement bias, and confounding factors. Mendelian randomization studies were scrutinized for the validity of the genetic instruments used and for adherence to the core assumptions underlying causal inference. Risk of bias was further considered in terms of the selection of study populations, the consistency and appropriateness of outcome measures, the completeness of reporting, and the potential influence of funding or conflicts of interest. The interpretation of study findings was informed by these assessments, highlighting areas supported by robust evidence while also acknowledging limitations arising from heterogeneity, methodological constraints, or potential biases. Publication bias was not formally assessed due to the small number of studies and the diversity of study designs included in this review. The overall certainty of evidence was not formally assessed using GRADE, but study quality and risk of bias were considered when interpreting the findings.

### 2.3. Review and Theoretical Frameworks

Recent reviews have established a comprehensive theoretical framework linking the microbiome to the development and progression of GBM and high-grade gliomas (HGGs). Keane et al. [8] offered a critical synthesis highlighting the multifaceted ways in which gut dysbiosis contributes to tumour biology. The authors emphasized that alterations in microbial communities impacted immune suppression, particularly through the regulation of regulatory T cells and glioma-associated macrophages/microglia (GAMs) with the critical role of microbiota-derived SCFAs in maintaining blood–brain barrier integrity. In the context of glioma, decreased SCFAs, especially butyrate levels were associated with impaired immune regulation and enhanced tumour immune evasion [8]. The authors also stressed that microbiome alterations can influence treatment efficacy. Dysbiosis can reduce responsiveness to standard therapies mainly by attenuating TMZ efficacy [8]. In parallel, Zhang et al. [9] focused specifically on the immunosuppressive microenvironment of GBM and its interaction with microbial factors. Their review underscored the importance of microbial modulation of immune checkpoints, particularly PD-1/PD-L1 and CTLA-4 molecules. They elucidated how microbiota-derived metabolites, mainly SCFAs ameliorate disease activity by contributing to decrease of proinflammatory Th1 and Th17 cells and increase of anti-inflammatory Tregs [9]. Nevertheless, the authors noted that other microbiome’s metabolite products, neurotransmitters: dopamine and serotonin may facilitate glial tumour growth and angiogenesis [9]. A conceptual framework was proposed by D’Alessandro et al. [10], in which microbial metabolites are positioned as key signalling mediators between the gut microbiota and the CNS, thereby influencing glioma progression through interactions with neural and immune components within the brain. Among these metabolites, various neurotransmitters have been identified as modulators of glioma behaviour. Dopamine has been reported to regulate cell survival and proliferation, and to enhance the growth of spheroids enriched in cancer stem cells. Serotonin has been shown to promote cell proliferation and migration, induce differentiation, and increase the release of the neurotrophic factor GDNF. Norepinephrine has been found to modulate proliferation while inhibiting migration and invasion. GABA has been associated with reduced proliferation and the maintenance of cellular quiescence. Glutamate has been implicated in the regulation of cell growth, enhancement of proliferation and migration, promotion of perivascular invasion, and overall stimulation of tumor growth and invasiveness [10]. Regarding neuroactive microbial metabolites, it has been suggested that SCFAs, in addition to inducing neuropeptide release by enteroendocrine cells, may also reach the brain by crossing BBB, potentially via monocarboxylate transporters (MCTs) abundantly expressed on endothelial cells and, upon entry, activate microglial cells. In relation to bacterial taxa, it was concluded that glioma induces specific alterations in the gut microbiota, characterized by a reduced *Firmicutes*:*Bacteroidetes* ratio and an increased relative abundance of the phylum *Verrucomicrobia*. This shift includes an elevation in the genus *Akkermansia*, particularly its predominant species *Akkermansia muciniphila* [10]. All reviews converged on the central idea that the gut microbiome acts as a pivotal regulator of glioma immune microenvironment, epigenetics and angiogenesis through numerous metabolic and signalling pathways [8,9,10]. They collectively proposed that restoring microbial diversity, enhancing beneficial metabolite production, microbiome profiling and targeting microbiota-influenced immune deviation could offer promising new strategies for glioma therapy [8,9].

### 2.4. Multi-Modal Mechanistic Studies

Emerging studies have provided mechanistic evidence linking microbial signals to immune modulation within GBM, revealing complex interactions that may underlie tumour immune evasion and progression. These findings illustrate how microbial presence, antigenicity, and metabolic influence can shape the glioma microenvironment. The ability of tumour-infiltrating lymphocytes (TILs) in GBM to recognize and respond to microbial peptides presented via HLA class II molecules was demonstrated by Naghavian et al. [11]. In their study, nearly 58% of the highly reactive peptides identified were found to be of bacterial origin. This suggests that microbial antigenic mimicry may be leveraged by gliomas to modulate anti-tumour immune responses. This mimicry could lead to either activation or tolerization of T cells, potentially skewing immune surveillance mechanisms in favour of tumour growth. Hence, TIL-derived CD4+ T cell clones (TCCs) cross-recognize glioma antigens as well as microbiota-derived peptides, which primarily originate from the phyla *Firmicutes*, *Proteobacteria* and *Bacteroidetes*. The study thus proposed a direct link between bacterial antigens and the functional state of TILs within the GBM microenvironment [11]. Building on the theme of bacterial presence within tumours, various types of human neoplasm, including glioblastoma multiforme, were examined by Nejman et al. [12]. This investigation was conducted using a combination of several techniques, among others, 16S rRNA gene sequencing, immunohistochemistry (IHC), and fluorescence in situ hybridization (FISH) for intratumoral microbial marks. It was reported that the noted bacteria primarily belonged to the phyla *Proteobacteria* and *Firmicutes*. Intracellular bacteria were also observed within tumour-associated immune cells, suggesting a functional interaction between microbial components and host immunity [12]. Expanding on a multi-omics study approach regarding microbial influence on neoplastic processes in the brain, the presence and location of intratumoral bacterial communities in GBM were identified and characterized by Li et al. [13]. This was accomplished through the employment of techniques including, among others, 16S rRNA sequencing combined with transcriptomics, metabolomics, IHC, multicolour immunofluorescence, and FISH. Furthermore, animal model research was applied to verify the impact of key bacterial phyla. In glioblastoma tissues, compared to adjacent unchanged brain tissues six bacterial genera were found in increased abundance: *Fusobacterium*, *Longibaculum*, *Pasteurella*, *Intestinimonas*, *Arthrobacter* and *Limosilactobacillus*. Moreover, their findings indicated that species *Fusobacterium nucleatum* was associated with enhanced tumour progression and increased expression of pro-inflammatory chemokines CXCL1, CXCL2 and CCL2. In addition, it was suggested in further analysis that the expression of neuron-related genes may be affected by microbial metabolites [13]. Collectively, a multifaceted landscape in which microbial elements are observed to actively interact with both immune and tumour cells within the glioma microenvironment is revealed by these studies. Through mechanisms involving antigenic mimicry, intracellular colonization, and metabolic reprogramming, the microbiome appears to contribute to glioma immune evasion and progression. Understanding these interactions may uncover novel therapeutic opportunities targeting microbe-immune-tumour dynamics.

### 2.5. Murine Model Studies

Experimental models using antibiotic-induced dysbiosis have provided valuable insights into how microbiota alterations can influence glioma development and progression. In murine models, specific changes in microbial communities have been linked to enhanced tumour growth, altered vascularization, and impaired immune surveillance. Data extraction was performed independently by two reviewers using a standardized form capturing study design, animal strain, interventions, microbiota changes, metabolite profiles, immune cell populations, tumor volume, and vascular markers. Any discrepancies were resolved via discussion. Rosito et al. [14] demonstrated that the administration of a combined antibiotic regimen consisting of vancomycin and gentamicin to glioma-bearing mice resulted in pronounced dysbiosis. This dysbiosis state was subsequently associated with antibiotic-induced impairment of microglial function and an enhancement of glioma growth. Furthermore, it was asserted that microbial imbalance facilitates the development of novel vascular networks essential for tumour expansion. The expression of CD34+, a key endothelial marker implicated in glioma progression, was examined. Elevated CD34 immunoreactivity within glioma tissues from antibiotic-treated mice was observed. Additionally, an increased presence of CD34+ vessel-like structures was reported in comparison to control mice, thereby indicating enhanced vasculogenesis [14]. Moreover, a shift in the levels of specific brain metabolites was indicated by the authors, including a reduction in the abundance of short-chain fatty acids (SCFAs) and a marked increase in glycine concentrations. Glycine was described as a promoter of angiogenesis and cellular stemness and was identified as a mediator contributing to glioma progression in antibiotic-treated mice [14]. Similarly, the effects of prolonged antibiotic exposure on innate immune responses in a murine glioma model were investigated by D’Alessandro et al. [15]. A disruption of the gut microbiota was observed in antibiotic-treated mice. This disruption was characterized by an increased abundance of the *Alcaligenaceae* and *Burkholderiaceae* families, along with a decreased presence of the *Prevotellaceae*, *Rikenellaceae*, and *Helicobacteraceae* families. Due to heterogeneity in outcomes and study designs, a narrative synthesis was performed. Effect measures included tumor volume (mm^3^), immune cell percentages (% of total), and metabolite concentrations (µM). These microbial shifts were found to impair the maturation of cytotoxic natural killer (NK) cells infiltrating brain tissue. In particular, a reduction in the mature CD27^+^/CD11b^+^ NK cell subset, recognized for its role in tumour surveillance, was reported. This immune dysfunction was correlated with an acceleration of glioma progression in mice exposed to antibiotics [15]. Remarkably, a reduction in tumour growth was observed following the interruption of antibiotic treatment. This effect was attributed to a partial restoration of the gut microbiota and a reversal of alterations within the NK cell subsets [15]. Collectively, these murine studies provide strong evidence that dysbiosis induced by antibiotic administration may promote glioma progression. This effect appears to be mediated through both the enhancement of tumour-supportive vasculature and the suppression of anti-tumour immune functions.

### 2.6. Cross-Species Research Projects

The intricate relationship between gut microbiota and glioma progression is increasingly understood through the lens of metabolomic alterations. Modifications in microbial composition alter metabolite profiles, thereby affecting tumour behaviour, immune regulation, and treatment responses in animal and human models. Among these metabolites, short-chain fatty acids (SCFAs), produced almost exclusively by gut bacteria, are recognized for their immunomodulatory properties. The importance of SCFAs in the microbiome–brain relationship is reflected in their influence on immune responses, including neutrophil chemotaxis, induction of regulatory T cells, enhancement of IL-10 secretion, inhibition of NF-κB signalling, and suppression of proinflammatory cytokine production by myeloid cells [16]. The microbial metabolic consequences of glioma development were examined by Dono et al. in faecal samples obtained from glioma-bearing mice and human GBM patients [16]. The methodology can be summarized as follows. Liquid chromatography–mass spectrometry and 16S rRNA sequencing were performed on faecal samples to evaluate metabolite levels and taxa abundance in both mice and humans. Data extraction included taxa relative abundance, metabolite concentrations, clinical characteristics (age, sex, tumor grade, IDH status), and treatment regimen. Two reviewers independently extracted and cross-verified all data. In the murine arm, GL261 cells, representing an invasive yet nonmetastatic glioma model with a high tumour take rate and harbouring both p53 and K-ras mutations, were implanted with or without temozolomide treatment. In the human arm, faecal samples from glioma patients were compared with those from healthy controls. In the human cohort, faecal samples obtained prior to surgical resection from newly diagnosed glioma patients, along with matched controls, were analysed using metabolomic and 16S rRNA sequencing, with comprehensive clinical, pathological, and molecular data [16]. In the murine model, alterations in both the composition of the microbiota and the concentrations of microbial metabolites were observed. A comparative analysis of faecal samples collected at baseline and from glioma-bearing mice revealed differences in taxa relative abundance, including decrease in *Bacteroidetes* and *Firmicutes* phyla levels and increased levels of the *Verrucomicrobia* phylum. An elevated relative abundance of the genera *Akkermansia* and *Bacteroides* was also noted following tumour growth in mice. Notably, in the presence of temozolomide, glioma-induced changes in the microbiome were attenuated [16]. Respectively, microbial faecal metabolites were detected at altered concentrations. Metabolites with increase subsequent to tumour growth included acetylcholine, serotonin, caproate and 3-methyl valerate. Opposite, glioma growth was associated with reduced levels of adenosine, histamine, butyrate, propionate, acetate, GABA, tryptophan, dihydroxy phenyl acetic acid (DOPAC), norepinephrine, 5-hydroxyindoleacetic acid (5-HIAA), valerate and aspartic acid [16]. However, no significant differences in the relative abundance of faecal bacterial taxa were observed between glioma patients and controls. Analysis of faecal metabolites in glioma patients, as compared to healthy individuals, revealed a reduction in the levels of 5-HIAA and norepinephrine [16]. Patrizz et al. [17] further expanded upon these findings through the analysis of faecal microbiota composition by 16S rRNA sequencing in both glioma-bearing mice and human patients compared to controls. In the experimental procedure, C57BL/6 mice were implanted with GL261 or sham tumours and subsequently treated with temozolomide (TMZ) or saline. The faecal samples were collected longitudinally and subjected to 16S rRNA sequencing analysis. Additionally, faecal samples were obtained from healthy controls as well as glioma patients at diagnosis, both prior to and following chemoradiation therapy [17]. An increase in the *Verrucomicrobia* phylum and *Akkermansia* genus, as well as a decrease in the F/B ratio, has been demonstrated in glioma patients and mice compared to healthy controls [17]. Furthermore, dysbiotic differences among patients with various glioma subtypes have been observed. A diverse increase in the abundance of bacterial phyla *Bacteroidetes* and *Verrucomicrobia* was reported between IDH-wildtype and IDH-mutant patients relative to controls, alongside an elevated presence of the *Akkermansiaceae* family and *Akkermansia* genus in the IDH-wildtype glioma group compared to the IDH-mutant group [17]. Likewise, it was noted that dysbiosis signatures present in glioma-developing mice were absent following temozolomide (TMZ) treatment. However, an analogous reversal effect was not detected in humans [17]. Collectively, these findings emphasize that systemic metabolic homeostasis is disrupted by glioma through alterations in the microbiota, and that therapeutic interventions further complicate this relationship. Targeting microbiota-derived metabolites, such as SCFAs, or preserving microbial diversity during treatment could represent promising adjunctive strategies to mitigate immune suppression and improve therapeutic outcomes. However, conclusions should be drawn cautiously due to the limited scope of available data from both animal models and human studies.

### 2.7. Therapy-Modulating Potential in Preclinical Models

Recent studies have highlighted the role of gut microbiota composition in influencing the efficacy of gliomas therapies, particularly adjunctive probiotic strategies and immunotherapy, among which virotherapy is an emerging branch. Modulation of the gut ecosystem has been emerged as a potential factor affecting treatment response. Data extraction included intervention type, microbiota composition (16S rRNA sequencing), treatment response (tumor volume, survival), immune markers (TILs, cytokines), and molecular pathways (e.g., PI3K/AKT, Ki-67). Meléndez-Vázquez et al. [18] investigated the relationship between gut microbiota and the efficacy of viral immunotherapy using the oncolytic virus *Delta-24-RGDOX*, which refers to the *Delta-24-RGD* virus employed in clinical therapy trials and enriched with the immune costimulatory ligand OX40 [18]. Microbiota profiling was performed through 16S rRNA gene sequencing on faecal samples from diverse groups of mice, including native mice without glioma, a control group with glioma without treatment, and groups subjected to various immunotherapy regimens involving *Delta-24-RGDOX* [18]. It was demonstrated that viroimmunotherapy-treated subjects exhibiting prolonged survival were characterized by increased abundance of the phyla *Verrucomicrobia* and *Actinobacteria*, as well as the genera *Akkermansia* and *Bifidobacterium*, respectively [18]. Likewise, it was reported that mice receiving any immunomodulatory treatment presented an overall microbiota profile, defined by the *Firmicutes*:*Bacteroidetes* ratio, more closely resembling that of the native tumour-free group than that of the untreated control group [18]. The direct administration of probiotics, specifically species *Bifidobacterium lactis* and *Lactobacillus plantarum*, was investigated by Wang et al. [19] to assess their effects on glioma progression in the GL261 orthotopic mouse model, in which GL261 cells—a murine glioma cell line—were implanted into the brain to reproduce tumour growth. Effect measures included tumor volume (mm^3^), survival time (days), relative abundance of bacterial taxa (%), and protein expression levels. Narrative synthesis was applied due to diverse interventions and outcome measures, with risk of bias evaluated per study design. Mice were randomly assigned to three groups: sham and model (Mod) groups, serving as controls, and the Mod-*L. plantarum* + *B. lactis* group, which constituted the experimental cohort receiving daily oral gavage of the probiotics. Stereotaxic injection of GL261 cells was performed in the Mod and Mod-*L. plantarum* + *B. lactis* groups, while sham mice received phosphate-buffered saline (PBS). Mice receiving probiotic treatment were found to have prolonged survival, and glioma-bearing mice in probiotic-treated group exhibited a significant reduction in tumour growth compared with the untreated Mod group, as determined by cerebral T2-weighted magnetic resonance imaging (MRI) [19]. A reduced abundance of the genus *Lactobacillus* was observed in glioma-bearing mice relative to healthy counterparts. Following probiotic administration, *Lactobacillus* genus levels were markedly increased, and this change was associated with a decreased abundance of potentially pathogenic bacteria, including genera *Helicobacter* and *Staphylococcus* [19]. Mechanistically, probiotic supplementation was found to restore tight junction proteins, such as Occludin, in the gut epithelium, indicating an improvement in intestinal barrier integrity. In addition, the treatment was shown to suppress the PI3K/AKT signalling pathway—a key driver of glioma cell proliferation and survival—and to downregulate the expression of the proliferation marker Ki-67 [19]. Collectively, these findings provide compelling evidence that modulation of the gut microbiota, through either therapeutic interventions or probiotic supplementation, can substantially influence the efficacy of glioma treatment. Targeting the microbiome is therefore regarded as a promising adjunctive strategy to enhance anti-glioma immune responses and to increase tumour sensitivity to both novel and existing therapies.

### 2.8. Human Cohort Studies

Translational research efforts have begun to extend findings from preclinical models into clinical contexts, exploring how gut and oral microbiota profiles might serve as biomarkers for glioma diagnosis, stratification, and prognosis. Recent human studies offer preliminary but promising evidence linking microbial patterns to tumour characteristics. Jiang et al. [20] conducted a pilot case–control study comparing the gut microbiota of glioma patients to those with benign meningiomas and healthy individuals. The results of faecal samples 16S rRNA gene sequence analysis showed that the diversity of the microbial ecosystem in both brain-tumour patient groups were less and related to lacked SCFA-producing bacteria, compared to the healthy controls. Microbial composition differences appeared in form of increased abundance of pathogenic bacteria like *Enterobacteriaceae* family in meningioma group, as well as overrepresentation of certain carcinogenic bacteria including genera *Fusobacterium* and *Akkermansia* in the glioma group [20]. A significant decrease in abundance was observed for members of the *Lachnospiraceae* family in glioma patients compared to healthy controls. Conversely, the *Bacteroidaceae* family exhibited the most substantial increase in occurrence within the glioma cohort compared to the healthy individuals. On genus level the numbers of *Bacteroides* and *Prevotella* was most divergent between tumour and control groups with substantially higher abundance in glioma patients [20]. Moreover, compared with healthy controls, brain tumour cohorts and particularly the glioma group displayed reduced metabolism in several pathways, including D-glutamine and D-glutamate, which participate in the tricarboxylic acid (TCA) cycle. Disruptions in these pathways, linked to gut dysbiosis, may contribute to tumour pathogenesis and progression [20]. In a complementary approach, Wen et al. [21] examined the oral microbiota across different glioma types, stratified by malignancy grade, low-grade gliomas (LGG, WHO grades I–II) and high-grade gliomas (HGG, WHO grades III–IV), as well as by the presence of isocitrate dehydrogenase 1 (IDH1) mutation. This mutation is frequently detected in IDH1-mutant gliomas, which typically correspond to LGG, but is rare in IDH1–wild-type gliomas and especially uncommon in grade IV tumours [21]. Data extraction captured patient demographics, tumor characteristics (grade, IDH1 status), microbiota composition (relative abundance), metabolite levels, and clinical interventions. Saliva samples 16S rRNA sequencing analysis revealed oral microbiome composition shifts between groups of HGG patients, LGG patients and healthy control. The genera *Capnocytophaga* and *Leptotrichia* were found to be inversely associated with glioma grade. Furthermore, five microbiota taxa were identified with a marked decrease in the HGG group compared with the healthy control group, which were genera *Porphyromonas*, *Haemophilus*, *Leptotrichia*, *Capnocytophaga* and species *TM7x* [21]. A significantly lower abundance of *Firmicutes* at the phylum level was observed in IDH-mutant samples compared to IDH–wild-type samples. At the genus level, *Bergeyella* and *Capnocytophaga* were found to be significantly positively associated with the IDH-mutant group [21]. Collectively, these human studies have shown that faecal and oral microbiome characteristics are associated with brain tumour malignancy and IDH1 mutation in glioma, indicating a potential role for the microbiota as a valuable tool in the diagnosis and prognostic stratification of brain gliomas.

### 2.9. Human Mendelian Randomization Studies

The three Mendelian randomization studies discussed are based on data from the MiBioGen consortium, which is considered the most extensive genome-wide meta-analysis of the human gut microbiome to date. This resource comprises 16S rRNA gene sequencing and host genotyping data from 18,340 individuals across over 20 European and international cohorts [22,23,24]. Mendelian randomization is a methodology that uses genetic variations as instrumental variables to establish whether a causal link exists between an exposure and a predetermined outcome. In the context of these studies, the methodology was applied to investigate potential causal links between gut microbial taxa and glioma risk. Specifically, single-nucleotide polymorphisms were utilized as instrumental variables, having been derived from genome-wide association studies [22,23]. By leveraging the harmonized MiBioGen dataset, these analyses aimed to move beyond mere correlations and identify specific bacterial genera associated with either increased susceptibility or protective effects. The use of Mendelian randomization allows for the minimization of confounding factors, leading to more reliable causal inferences [22,23]. Although differing in statistical approaches, the studies converge on the premise that genetic interactions between host and microbiome may contribute to glioma pathogenesis [22,23,24]. A two-sample Mendelian randomization analysis was performed by Cui et al. [22] to explore the causal link between specific bacterial populations composing the gut microbiome and the susceptibility to glioma. Inverse variance weighting (IVW) was used as the primary method for obtaining causal estimates, while Mendelian randomization-Egger and weighted median models were employed to ensure robustness. To account for pleiotropy, sensitivity testing was also conducted, including leave-one-out and heterogeneity analyses [22]. Eight taxa were identified with suggestive associations. At the genus level, *Adlercreutzia*, *Prevotella7* and *Catenibacterium* were associated with increased glioma risk, while *Coprobacter* and *Olsenella* showed protective associations. At the family level, *Peptostreptococcaceae* was inversely associated with glioma, indicating a potentially protective effect. Two phylum-level taxa, *Verrucomicrobia* and *Euryarchaeota*, were also linked to increased glioma risk [22]. A cautious interpretation of the findings was warranted, as not all associations passed the False Discovery Rate (FDR) correction. However, the biological probability of these reciprocities within gut–brain axis models was emphasized. It was speculated that risk-associated taxa promote neuroinflammation. Conversely, these processes may be counteracted by protective taxa via immune regulation through the modulation of CTLA-4 or PD-1 and their ligand PD-L1 [22]. Mendelian randomization study was also conducted by Wang et al. [23] to investigate causal associations between gut microbiota and GBM. To be precise, Mendelian randomization-Egger, IVW, and weighted median models were employed, with further validation being performed through Mendelian randomization-PRESSO and heterogeneity analyses [23]. A total of nine microbial taxa were found to be associated with GBM. Among them, the genus *Eubacterium brachy group* and the family *Peptostreptococcaceae* were identified as risk-enhancing taxa. Protective associations were identified for six genera: *Anaerostipes*, *Faecalibacterium*, *Lachnospiraceae UCG004*, *Phascolarctobacterium*, *Prevotella7*, and *Streptococcus*, as well as for the family *Ruminococcaceae*. It is notable that a statistically significant protective effect was retained solely by *Ruminococcaceae* after FDR correction [23]. A further finding was that the protective genera were largely characterized as SCFA producers, particularly of butyrate, which may inhibit pro-tumour immune pathways and support anti-inflammatory responses [23]. A potential causal role of specific gut microbes in GBM etiopathogenesis was underlined by the authors. This was further reinforced by an inverse Mendelian Randomization analysis, a method used to evaluate a reverse causal relationship, which did not support a causal influence of glioblastoma on altered gut microbiota [23]. Consequently, a more likely direction of the relationship, with the microbiota affecting the development of GBM and not the other way around, was concluded by the authors [23]. A bidirectional Mendelian randomization study, applying an analogous methodology, was also conducted by Zeng et al. [24], and in it, several microbial taxa were identified as potentially causally linked to glioblastoma. Data extraction included bacterial taxa at genus, family, and phylum levels, effect sizes (odds ratios with 95% CI), *p*-values, FDR correction, and MR model parameters. Two independent reviewers verified extracted data. Among those associated with increased risk, the most prominent findings involved the *Eubacterium brachy group* genus and the *Peptostreptococcaceae* family [24]. Conversely, the strongest protective relationships were observed for the genera *Anaerostipes* and *Faecalibacterium*, as well as the *Ruminococcaceae* family [24]. An association with causal significance in both directions was identified between *Prevotella7* genus and glioblastoma, as supported by both forward and reverse Mendelian randomization analyses. This dual relationship was interpreted to suggest that *Prevotella7* may have diagnostic and therapeutic relevance in the context of glioblastoma [24]. Effect measures were expressed as odds ratios (OR) for glioma per unit change in bacterial abundance. Narrative synthesis was conducted due to variability in taxa and MR models. Nevertheless, while Mendelian randomization studies offer powerful insights, they also have limitations, including a reliance on genetic proxies, the assumption of no pleiotropy, and an inability to capture environmental influences. Therefore, future studies combining Mendelian randomization with longitudinal clinical and microbiome datasets are needed to fully elucidate the causal pathways linking gut microbes to glioma biology.

## 3. Results

The literature search identified 56 records through database searching, all retrieved from MEDLINE, with no additional records identified from the Cochrane database or other sources. All identified records were screened at the title and abstract level, resulting in the exclusion of 33 records. Subsequently, 23 full-text articles were assessed for eligibility. Following full-text evaluation, 6 articles were excluded due to duplication, insufficient data, or failure to meet the predefined exclusion criteria. Ultimately, 17 studies were included in the qualitative synthesis. The study selection process is illustrated in the PRISMA 2020 flow diagram (Figure 1). Details of the search strategy, eligibility criteria, and screening procedures are described in Section 2.1.

To facilitate a visual consolidation of the key characteristics across all the analyzed literature, a tabular summary detailing study designs, primary biological factors captured, and briefly summarized outcomes was provided in Table 1 (‘Studies included in the systematic review’). The heterogeneity of the analyzed studies should be noted. Considering the divergences in research models, populations, interventions, and outcome measures, the application of quantitative metrics and formal subgroup or sensitivity analyses were not feasible due to the narrative synthesis approach. A comprehensive synthesis of the reviewed literature highlights several microbial taxa and associated metabolites that recur across murine, human, and genomic research contexts, indicating biologically plausible and translationally relevant mechanisms in glioma pathophysiology. Acknowledging that visual representation will enhance the comprehension of the subsequent descriptive content, we provide below tabular summaries illustrating the principal correlations (Table 2 and Table 3). Overall, the included studies comprised experimental murine models, human observational and case–control studies, Mendelian randomization analyses, multi-omics investigations, and narrative reviews, as summarized in Table 1.

Risk of bias assessment indicated an overall moderate level of bias across the included studies. Preclinical murine experiments were primarily limited by small sample sizes and model-specific constraints, while human observational studies were susceptible to confounding and selection bias. Mendelian randomization studies were considered less prone to confounding, although potential pleiotropy and instrument validity limitations were noted.

Due to substantial heterogeneity in study designs, populations, outcome definitions, and analytical approaches, quantitative effect estimates and summary statistics were not consistently available. Therefore, individual study results are presented narratively, emphasizing the direction and biological relevance of reported associations.

### 3.1. Recurrent Microbial Signatures and Metabolic Pathways

Recurrent patterns of microbial dysbiosis associated with glioma have been identified across both human and murine models. Analysis at the phylum level consistently reveals a reduced *Firmicutes*:*Bacteroidetes* ratio, which serves as a major hallmark of microbiome imbalance driven by decreased *Firmicutes* phylum and elevated *Bacteroidetes* phylum [10,17]. The peculiarity of *Firmicutes* concerns its significance and yet inconclusiveness. While a protective role is often expected due to its reduction in glioma individuals [16,17], a higher abundance of this phylum was paradoxically noted in patients with worse prognosis IDH–wild-type glioblastoma compared to IDH-mutant individuals [21]. This duality suggests that the clinical significance of *Firmicutes* may be contingent upon specific glioma subtypes or the precise genera being examined. The microbial components of the *Firmicutes* phylum, along with those of *Proteobacteria* phylum, are thought to be present intratumorally and influence the immune microenvironment of glioma [11,12]. The phylum *Verrucomicrobia* is also consistently reported taxon, which has been overrepresented in individuals with glioma. Within this phylum, the genus *Akkermansia*, particularly *Akkermansia muciniphila*, is the most consistently reported taxon and is found in increased abundance in both murine and human glioma samples [10,16,17,22]. Its elevated presence is often accompanied by reduced SCFA concentrations and impaired neurotransmitter levels, concurrent with immune dysregulation that is prone to neuroinflammation [10,16,17,22]. Significant importance is also attributed to the aforementioned phylum *Bacteroidetes* in the context of glioma [17]. An influence on the evasion of immune surveillance by glioma cells is attributed to its microbial peptides, which are cross recognized by T cells in a manner analogous to tumour antigens [11]. Among this phylum’s constituent taxa, attention should be paid especially to the *Bacteroidaceae* family and the genus *Bacteroides*, which exhibits a significant association with glioma growth and was distinguished by the highest increase in the glioma group compared to healthy individuals [16,20]. Furthermore, a significant finding concerning intratumoral bacterial particles relates to the phylum *Fusobacteriota*. Specifically, the detection of the species *Fusobacterium nucleatum*, a constituent of this phylum, has been linked to increased expression of pro-inflammatory chemokines such as CXCL1, CXCL2, and CCL2, alongside enhanced angiogenesis and tumour gene expression alterations [13,20]. As previously discussed, a protective role is attributed to the phylum *Firmicutes*, specifically the genus *Lactobacillus*, which belongs to this phylum. The second taxon whose protective role is frequently emphasized is the representative of the phylum *Actinobacteria*, the genus *Bifidobacterium*. Both genera have been associated with beneficial effects in murine models. The abundance of these genera was increased in response to probiotic supplementation and virotherapy, respectively, resulting in enhanced gut barrier integrity, inhibition of the PI3K/AKT pathway, and prolonged survival [18,19]. The involvement of these genera in the restoration of intestinal tight junctions and the reduction of tumour-associated markers such as Ki-67 has also been demonstrated, which may indicate not only their protective function but also their therapeutic potential [19]. Furthermore, protective associations are indicated by Mendelian Randomization studies for SCFA-producing taxa, including the family *Ruminococcaceae*, especially its representative genus *Faecalibacterium*. Conversely, increased glioma risk was associated with the genera *Adlercreutzia* and *Catenibacterium*, as well as species of the *Eubacterium brachy group* [22,23,24]. Noteworthily, the genus *Prevotella7* presents an inconsistent and complex character, as it is simultaneously linked to increased glioma risk [22] and identified as a protective genus in subsequent analyses [23]. A reciprocal relationship between *Prevotella7* and glioblastoma, validated by both forward and reverse Mendelian Randomization analyses, suggests a bidirectional causal connection [24]. The identified dysbiosis patterns are intrinsically linked to corresponding metabolic disturbances. Concurrent metabolic findings include reduced concentrations of SCFAs, GABA, tryptophan, 5-HIAA, and norepinephrine [14,16], which are critical for immune modulation, blood–brain barrier integrity, and anti-inflammatory homeostasis [8,9,10]. Additional metabolic disturbances associated with dysbiosis include elevated levels of glycine, serotonin, glutamate, dopamine, and acetylcholine molecules known to be involved in angiogenesis, glioma stemness, and tumour proliferation [10,14,16]. Moreover, murine studies have demonstrated that antibiotic-induced depletion of the gut microbiota impairs innate anti-tumour immunity. Specifically, a decline in the maturation of CD27^+^/CD11b^+^ natural killer (NK) cells was observed, accompanied by accelerated glioma progression. Importantly, these immune and tumour-related abnormalities were shown to be reversible following the cessation of antibiotic exposure [15]. The qualitative synthesis revealed consistent directional associations across studies, despite considerable heterogeneity in experimental models, host species, microbiome sampling sites, and glioma subtypes. Increased abundance of pro-inflammatory or intratumoral taxa was generally associated with tumour progression, whereas SCFA-producing and probiotic-associated taxa were linked to protective or therapeutic effects. Heterogeneity was largely attributable to differences in IDH mutation status, tumour grade, and analytical methodologies. Sensitivity analyses were not conducted due to the qualitative nature of the synthesis and the limited number of studies within individual methodological subgroups.

### 3.2. Translational Relevance

A subset of studies points toward potentially clinically relevant findings regarding glioma-associated microbiome alterations. Viroimmunotherapy using *Delta-24-RGDOX* was more effective in mice with microbiota characterized by elevated abundance of the *Akkermansia* and *Bifidobacterium* genera [18]. Furthermore, dysbiotic differences among patients with various glioma subtypes have been observed. An elevated presence of the *Akkermansiaceae* family and *Akkermansia* genus in the IDH-wildtype glioma group compared to the IDH-mutant group was demonstrated [17]. Moreover, oral human microbiota composition differs significantly by glioma grade and IDH1 mutation status. Genera such as *Capnocytophaga* and *Leptotrichia* were more prevalent in low-grade gliomas and IDH1-mutant tumours, indicating possible diagnostic or prognostic value [21]. These findings highlight that gut microbiome composition may not only reflect disease status but may also potentially serve as prognostic markers and affect the response to emerging therapies. Assessment of reporting bias was limited by the predominance of positive findings and the lack of registered protocols among the included studies, suggesting a potential risk of publication and selective reporting bias. Overall, the certainty of evidence was considered low to moderate, reflecting reliance on preclinical models, observational human studies, and Mendelian randomization analyses. Although biological plausibility was consistently supported, heterogeneity and methodological limitations reduce confidence in definitive causal inference.

## 4. Discussion

The synthesis of the preceding findings, derived from diverse methodologies including, among others, 16S rRNA gene sequencing in both murine and human cohorts, metabolomics, Mendelian Randomization studies, and the investigation of intratumoral bacterial communities, offers emerging insights into the involvement of the gut–brain axis in glioma pathogenesis. A notable consistency is reported regarding the pattern of dysbiotic shift in microbial composition, primarily characterized by a reduced *Firmicutes*:*Bacteroidetes* ratio [10,17]. This ratio, although a general metric of gut microbiome balance, is frequently utilised, and its deployment is substantiated across various models where a reduced value is interpreted as a marker of dysbiosis [10,17]. This broad pattern of microbial alteration provides the foundation for extending the consideration of mechanisms concerning the phylum *Bacteroidetes* in the context of increased risk of glioma development and progression. The pathogenic role of this phylum is substantially supported by evidence that glioma antigens and microbiota-derived peptides originating from the phylum *Bacteroidetes* are cross-recognized by T cell clones derived from tumour-infiltrating lymphocytes [11]. This microbial antigenic mimicry is hypothesised to modulate anti-tumour immune responses, resulting in the skewing of immune surveillance toward tumour advancement [11]. The family *Bacteroidaceae* and the genus *Bacteroides* exhibit a significant increase in occurrence within glioma patients, further underscoring the relevance [16,20]. A recurrent indication as a glioma risk factor is further the overabundance of the phylum *Verrucomicrobia*, a finding consistently highlighted across both human and murine samples [10,16,17,22]. Specifically, the genus *Akkermansia* and the species *Akkermansia muciniphila*, which belong to the phylum *Verrucomicrobia*, are found in increased abundance in both genetically engineered murine models of glioma and human patient samples [10,16,17,22]. Its elevated presence is frequently accompanied by a pro-neuroinflammatory state with impaired levels of crucial SCFAs and neurotransmitters [10,16,17,22]. The microbial influence is further demonstrated by the presence of intratumoral microbial components [12,13]. The dominant intratumoral phyla are *Proteobacteria* and *Fusobacteriota* [12]. The presence of these signatures is considered to functionally influence the tumour immune microenvironment [11,12]. This interaction is associated with an immunological response linked to enhanced angiogenesis and increased expression of pro-inflammatory chemokines, alongside alterations in tumour gene expression [13,20]. A more nuanced picture is revealed concerning the phylum *Firmicutes*, whose general depletion in glioma individuals suggests a protective effect [16,17]. However, contradictory findings link its higher abundance to the worse prognosis of IDH-wild-type glioblastoma [21]. This phylum is also observed to be among the intratumoral microbial components [12]. This duality underscores that the simple F/B ratio may be insufficient as a prognostic tool and that the clinical significance of the phylum *Firmicutes* may be contingent upon specific glioma subtypes or the precise genera of this phylum being examined. Nevertheless, a recurrent pattern of pro-neoplastic metabolic disturbances presented as concomitant with glioma-associated dysbiosis is the reduction in concentrations of SCFAs and key neurotransmitters, namely GABA, tryptophan, 5-HIAA, and norepinephrine. This reduction occurs concurrently with elevated levels of glycine, serotonin, glutamate, dopamine, and acetylcholine [10,14,16]. Additionally, antibiotic-induced dysbiosis was shown to accelerate glioma progression, due to impaired innate anti-tumour immunity, characterized by the disrupted maturation of CD27^+^/CD11b^+^ NK cells [15]. Furthermore, the genus *Eubacterium brachy group* has been indicated as a risk factor in Mendelian Randomization studies [22,23,24]. Notably, while the genus *Prevotella7* presents an ambiguous character, having been reported as both a risk factor [22] and a protective agent [23], a bidirectional causal connection with glioblastoma has been established via forward and reverse Mendelian Randomization analyses [24]. Consistently reported protective associations include SCFA-producing taxa, encompassing the genus *Anaerostipes*, as well as the entire family *Ruminococcaceae* with specific regard to its representative genus *Faecalibacterium* [22,23,24]. Protective associations are also reported for probiotic genera like *Lactobacillus* and *Bifidobacterium*. These genera have been shown to restore intestinal tight junctions, suppress the PI3K/AKT pathway, and reduce tumour markers such as Ki-67 [18,19]. Furthermore, viroimmunotherapy using *Delta-24-RGDOX* was demonstrated to be more effective in mice with microbiota characterized by elevated genera *Akkermansia* and *Bifidobacterium* abundance [18]. Furthermore, oral microbiota differences are associated with both glioma grade and IDH1 mutation status [21]. Regarding tumour malignancy, oral microbiota representative genera *Capnocytophaga* and *Leptotrichia* were found to be inversely associated with glioma grade. These findings suggest a potential role for the oral microbiome as a valuable tool in the prognostic stratification of brain gliomas [21]. Despite the convergence of many findings, several inconsistencies underscore the complexity of the relationship between microbiota and gliomas. The majority of mechanistic insights are derived from preclinical murine models [15]. Additionally, while the Mendelian Randomization studies provide preliminary evidence for a causal link, they have inherent limitations, whereby the etiopathogenesis must be elucidated through an integrated investigation [24]. The deployment of advanced computational algorithms, such as graph neural networks, is considered important for reliably identifying tumour-associated immuno-microbiome interactions. The integrated analysis of multi-omics data derived from single-cell and spatial sequencing technologies is also suggested as valuable to determine the functional role of microorganisms within the tumour microenvironment [25]. Moreover, future investigations using murine models of microbial influence on human disease should incorporate humanized gnotobiotic models. These are generated by the transplantation of specific human microbial communities into germ-free mice, thereby providing a powerful tool to faithfully mimic the human microbial system in vivo [26,27]. The development and application of standardized criteria for these models is deemed crucial for ensuring the accuracy and reproducibility of findings [26,28]. Despite these findings, the current body of evidence is limited by the small number of human studies, variability in experimental designs, and potential biases inherent to both preclinical models and Mendelian Randomization analyses. Methodological limitations of this review include the reliance on narrative synthesis due to heterogeneity of study designs and outcomes, and the absence of formal subgroup or sensitivity analyses. These findings highlight the potential of microbiota-targeted interventions, such as probiotics or microbiome-informed immunotherapies, as adjunct strategies in glioma management, and underscore the need for standardized protocols in both preclinical and clinical research.

## 5. Conclusions

The cumulative evidence underscores the intricate involvement of the gut–brain axis in glioma pathogenesis. The core microbial alteration is consistently defined by a reduced *Firmicutes*:*Bacteroidetes* ratio [10,17], which serves as a major hallmark of dysbiosis. The phylum *Bacteroidetes* and its constituent taxa often exhibit a pathogenic role, implicated in immune evasion through microbial antigenic mimicry that modulates anti-tumour immunity [11,16,20]. However, the significance of the phylum *Firmicutes* remains partially inconclusive. The intricate role of this phylum is highlighted by its demonstration of both protective associations, given its general depletion in glioma individuals, and a contradictory link, where findings associate its higher abundance in individuals with the worse prognosis of IDH-wild-type glioblastoma [16,17,21]. Consequently, this ambiguity highlights the visible limitations of the simple F/B ratio as a standalone prognostic tool. Regardless, the recurrently reported risk factor is the overabundance of the phylum *Verrucomicrobia*, specifically the genus *Akkermansia* [10,16,17,22], which is correlated with the reduction of crucial SCFAs and neurotransmitters, contributing to neuroinflammatory states, disrupted immune homeostasis, and impaired BBB integrity [8,9,10,16]. Likewise, the occurrence of intratumoral microbiota signatures is also relevant. The dominant taxa are the phyla *Proteobacteria* and *Fusobacteriota*, the presence of which is considered to functionally affect the glioma immune microenvironment [11,12,13]. Conversely, consistently reported protective associations are substantiated by Mendelian Randomization (MR) studies that suggest a causal link. These findings highlight the functional importance of SCFA-producing taxa, including the genera *Anaerostipes* and *Faecalibacterium* alongside the *Ruminococcaceae* family [22,23,24]. Furthermore, probiotic genera like *Lactobacillus* and *Bifidobacterium* exhibit protective roles by restoring gut barrier integrity, suppressing the PI3K/AKT pathway, and reducing tumor markers such as Ki-67 [18,19]. Furthermore, the protective significance extends to oral microbiota, specifically concerning the genera *Capnocytophaga* and *Leptotrichia*, inversely associated with glioma grade, suggesting a potential role in the prognostic stratification of gliomas [18,21,24]. Notwithstanding promising findings and considering the complexity of the role of the microbiome in glioma etiopathogenesis, as well as the limitations of current research methodologies, the necessity for further investigation must be underscored. Methodological progress should primarily focus on the deployment of upgraded standards of preclinical research, including gnotobiotic models in murine research, the application of advanced computational algorithms, and integrated multi-omics data analysis [25,26,27,28]. Crucially, this advancement must be paralleled by the expansion of longitudinal human cohort studies to ensure translational validity.

## Figures and Tables

**Figure 1 ijms-27-00444-f001:**
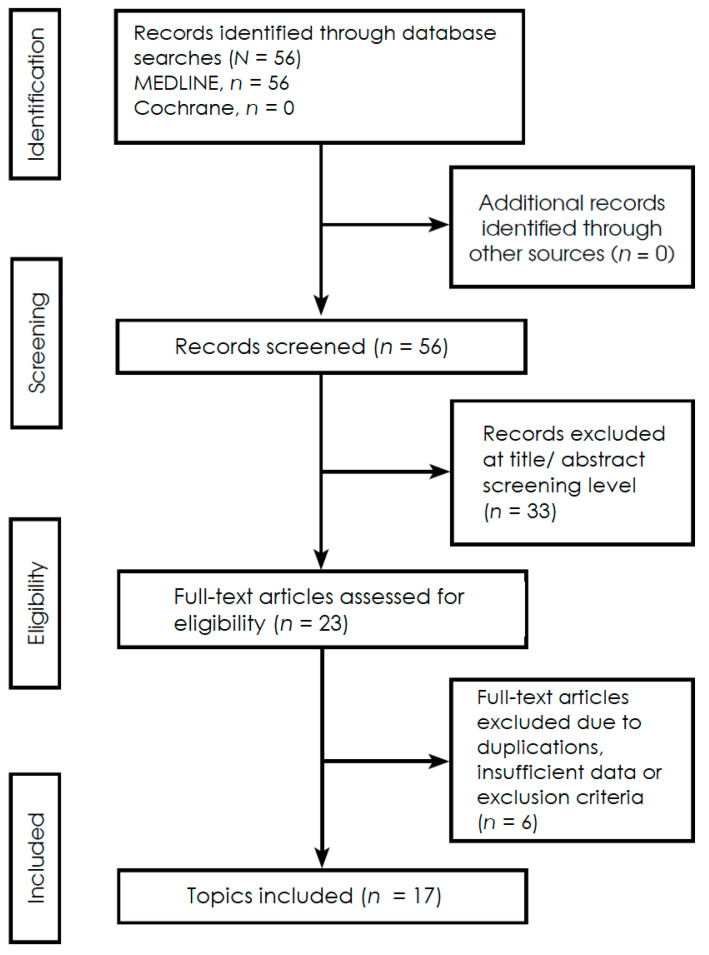
PRISMA 2020 flowchart illustrating the selection process of studies included in this systemic review (Appendix A).

**Table 1 ijms-27-00444-t001:** Studies included in the systematic review.

Author/Year	Characteristics	Key Biological Factors	Main Findings
Keane et al., 2025 [8]	Review; gut-brain axis in HGG pathology	Tregs, GAMs, SCFAs, butyrate, immune evasion, BBB integrity	The gut microbiome modulates the glioma microenvironment and may modulate anti-tumor immune responses and therapeutic outcomes
Li et al., 2025 [13]	Multi-omics; 50 human glioma patients and GL261 mouse model; investigating intratumoral bacteria	Genera in increased abundance: *Fusobacterium*, *Longibaculum*, *Pasteurella*, *Intestinimonas*, *Arthrobacter*, *Limosilactobacillus*; chemokines CXCL1, CXCL2 and CCL2; neuron-related genes expression	Several bacterial taxa exhibit increased intratumoral occurrence in glioma, notably Fusobacterium nucleatum, which promotes progression by increasing expression of pro-inflammatory chemokines altering the tumor transcriptome
Rosito et al., 2024 [14]	Experimental; Murine model study on antibiotic-induced dysbiosis in a GL261 mouse model of glioma	Expression of CD34+, reduced SCFAs and increased glycine concentrations, enhanced vasculogenesis	Dysbiosis promotes vasculogenesis and glioma stemness, shifting the tumor microenvironment toward a pro-angiogenic and pro-tumorigenic phenotype
Meléndez-Vázquez et al., 2024 [18]	Original Article; research on gut microbiota and the efficacy of Delta-24-RGDOX oncolytic virotherapy in mice	Prolonged survival in subjects with increased abundance of phyla *Verrucomicrobia*, *Actinobacteria*, and genera *Akkermansia*, *Bifidobacterium*	Gut microbiota composition influences virotherapy efficacy; specific taxa correlate with enhanced anti-tumor immune responses
Cui et al., 2024 [22]	Mendelian Randomization (MR) study exploring causal links between gut microbiota and glioma risk	Specific microbial taxa; neuroinflammation; modulation of CTLA-4, PD-1/PD-L1	Specific microbial shifts potentially have a causal impact on glioma risk via the gut-brain axis
Wang et al., 2024 [23]	Mendelian Randomization (MR) study investigating the association between gut microbiota and glioblastoma	Specific microbial taxa; SCFAs	Evidence of causal associations between specific microbial groups and the risk of developing glioblastoma
Naghavian et al., 2023 [11]	Experimental study; Human GBM tissues (*n* = 12) and T-cell clones	TILs, TCCs, HLA-presented bacterial peptides, phyla *Firmicutes*, *Proteobacteria*, *Bacteroidetes*	Glioblastoma infiltrating lymphocytes recognize microbial peptides presented via HLA-II molecules, suggesting that microbial antigenic mimicry may modulate anti-tumor immunity to favor tumor growth
Zeng et al., 2023 [24]	Bidirectional Mendelian Randomization (MR) study on gut microbiota and glioblastoma risk	Specific microbial taxa; both directions causal significance between *Prevotella7* genus and glioblastoma	Several microbial taxa show a causal impact on glioblastoma occurrence, reinforcing the role of the gut-brain axis in tumor development
Zhang et al., 2022 [9]	Review; immunosuppressive microenvironment in GBM	PD-1/PD-L1, CTLA-4, Th1 cells, Th17 cells, Tregs, GAMs, MDSCs; SCFAs, IL-10, TGF-β, microbial metabolites, tryptophan, neurotransmitters dopamine, serotonin	Gut microbiota modulates anti-tumor immune responses by regulating the GBM-induced immunosuppressive microenvironment and influencing tumor angiogenesis
Wang et al., 2022 [19]	Experimental study; original research on the effects of *B. lactis* and *L. plantarum* probiotics in glioma-bearing mice	Intestinal barrier integrity, tight junction proteins; the PI3K/AKT pathway; expression of Ki-67	Probiotic combination inhibits glioma growth by suppressing the PI3K/AKT signaling pathway and reinforcing the tightness of the intestinal barrier
Jiang et al., 2022 [20]	Human cohort study; original research comparing gut microbiota in patients with benign (*n* = 32) and malignant (*n* = 27) brain tumors	Specific bacterial abundance differences; metabolism pathways alternations: D-glutamine and D-glutamate	Reduced microbial diversity and malignancy-specific taxa in brain tumor patients suggest that gut dysbiosis-linked disruptions may contribute to tumor pathogenesis and progression
D’Alessandro et al., 2021 [10]	Review; neuro-signals from the gut microbiota and their role in glioma progression	Dopamine, serotonin, norepinephrine, GABA, glutamate; tumor-driven dysbiosis, reduced F:B ratio, increased *Verrucomicrobia* phylum	Microbial neuro-active signals may influence tumor progression, while the presence of glioma leads to glioma-induced dysbiosis
Wen et al., 2021 [21]	Human case-control study; original research investigating the link between oral microbiota and glioma grade; glioma patients (*n* = 70) and controls (*n* = 54)	Occurrence differences in phylum Firmicutes; genera Capnocytophaga, Leptotrichia, Porphyromonas, Haemophilus, Leptotrichia, Capnocytophaga, Bergeyella; species TM7x. LGG, HGG, IDH1 mutation	Oral microbial diversity and specific taxa composition correlate with the pathological grade of glioma
Nejman et al., 2020 [12]	Multi-omics; analysis of the intratumor microbiome in 1526 tumors, including brain tumors	Tumour-associated immune cells, intratumoral phyla *Proteobacteria*, *Firmicutes*	Diverse neoplasms, including brain tumors, possess distinct, predominantly intracellular microbiome, suggesting a functional interaction between microbial components and immune responses
D’Alessandro et al., 2020 [15]	Experimental; short communication; Murine model study investigating gut microbiota and innate immunity in antibiotic-treated GL261 glioma-bearing mice	Families in increased abundance: *Alcaligenaceae*, *Burkholderiaceae* and decreased abundance: *Prevotellaceae*, *Rikenellaceae*, *Helicobacteraceae*; reduction in the mature CD27+/CD11b+ NK cells; microglia	Antibiotic-induced dysbiosis impairs NK cell recruitment and shifts microglia toward a pro-tumor phenotype
Dono et al., 2020 [16]	Preliminary communication; cross-species project using GL261 mice and human glioma samples to investigate alternations in fecal metabolites and microbiome	Fecal microbial metabolites, SCFAs, neurotransmitters; in glioma-bearing mice decrease in phyla: *Bacteroidetes*, *Firmicutes* and increased in phylum *Verrucomicrobia*	Glioma development alters microbial metabolite profiles, affecting tumor behavior and immune responses in both animal and human models
Patrizz et al., 2020 [17]	Longitudinal study; cross-species research on alternations in gut microbiome in mice (GL261) and human glioma patients (*n* = 12)	In glioma patients and mice: an increase in the phylum *Verrucomicrobia*, genus Akkermansia, reduced F/B ratio; IDH1 mutation	Glioma development induces gut dysbiosis, exhibiting different mircobial shift signatures between IDH-wildtype and IDH-mutant subtypes

**Table 2 ijms-27-00444-t002:** Summary of microbial taxa identified in glioma patients and their associated role in glioma pathogenesis.

Taxonomy	Change in Numerosity	Related Effect	Research Methods and Source
*Firmicutes*	↓	Tumour growth	Murine model (mainly) and cross-species projects, stool examination [10]
*Bacteroidetes*	↑		
*Verrucomicrobia*	↑		
*Akkermansia muciniphila*	↑		
*Fusobacterium*, *Fusobacterium nucleatum*	↑ ↑	Enhanced tumour progression, increased expression of proinflammatory chemokines CXCL1, CXCL2 and CCL2	16S rRNA sequencing combined with transcriptomics, metabolomics, IHC, multicolour immunofluorescence, and FISH; animal models [13]
*Longibaculum*	↑		
*Pasteurella*	↑		
*Intestinimonas*	↑		
*Arthrobacter*	↑		
*Limosilactobacillus*	↑		
*Alcaligenaceae*	↑	Impaired maturation of cytotoxic natural killer (NK) cells infiltrating brain tissue, reduced mature CD27^+^/CD11b^+^ NK cell subset	Murine model, exposition to antibiotics [17]
*Burkholderiaceae*	↑		
*Prevotellaceae*	↓		
*Rikenellaceae*	↓		
*Helicobacteraceae*	↓		
*Lachnospiraceae*	↓	Mutation in IDH1-mutant gliomas, typically corresponding to LGG, rare in IDH1–wild-type gliomas, uncommon in grade IV tumours	Oral microbiota examination (saliva samples 16S rRNA sequencing analysis), stratified by malignancy grade, low-grade gliomas (LGG, WHO grades I–II) and high-grade gliomas (HGG, WHO grades III–IV), presence of IDH1 mutation [21]
*Enterobacteriaceae*	↑		
*Bacteroides*	↑		
*Prevotella*	↑		
*Capnocytophaga*	↓	Higher glioma grade	
*Leptotrichia*	↓		
*Porphyromonas*	↓		
*Haemophilus*	↓		
*Capnocytophaga*	↓		
*TM7x*	↓		
*Bergeyella*	↑	Presence of isocitrate dehydrogenase 1 (IDH1)	
*Capnocytophaga*	↑		
*Firmicutes*	↓		
*Adlercreutzia*	↑	Increased glioma risk	Human Mendelian randomization studies, 16S rRNA gene sequencing and host genotyping [22,23,24]
*Prevotella*	↑		
*Catenibacterium*	↑		
*Coprobacter*	↓		
*Olsenella*	↓		
*Peptostreptococcaceae*	↓		
*Verrucomicrobia*	↑		
*Euryarchaeota*	↑		
*Anaerostipes*	↓		
*Faecalibacterium*	↓		
*Lachnospiraceae UCG004*	↓		
*Phascolarctobacterium*	↓		
*Streptococcus*	↓		
*Ruminococcaceae*	↓		

**Table 3 ijms-27-00444-t003:** The GBA’s key microbial metabolites and neurotransmitters and their role in the glioma pathogenesis.

Molecule	Change in Concentration	Biological Role	Impact on Glioma Progression
Acetate	↑	Crossing the BBB, decrease of proinflammatory Th1 and Th17 cells and increase of anti-inflammatory Tregs, neutrophil chemotaxis, enhancement of IL-10 secretion, inhibition of NF-κB signalling, suppression of proinflammatory cytokine production by myeloid cells	↓
Propionate	↑		↓
Butyrate	↑		↓
GABA	↑	Reduced proliferation and maintenance of cellular quiescence	↓
Serotonin	↑	Promotion of cell proliferation and migration, induction if differentiation, increase in the release of GDNF	↑
Dopamine	↑	Regulation of cell survival and proliferation, enhancement of the growth of spheroids enriched in cancer stem cells	↑
Norepinephrine	↑	Modulation of proliferation and inhibition of migration and invasion, maintenance of BBB integration	↓
Acetylcholine	↑	Tumour growth, angiogenesis, glioma stemness, tumour proliferation	↑
Glutamate	↑	Cell growth, enhancement of proliferation and migration, promotion of perivascular invasion, overall stimulation of tumour growth and invasiveness	↑
Glycine	↑	Promotion of angiogenesis and cellular stemness, a mediator contributing to glioma progression (in antibiotic-treated mice)	↑
Tryptophan	↓	Gliomas exploit the kynurenine pathway to suppress anti-tumour immune responses via upregulation of IDO1 and TDO, modulated by microbial signals	↑

## Data Availability

The original contributions presented in this study are included in the article. Further inquiries can be directed to the corresponding author.

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
