# Peer review of "Emerging Insights into the Role of the Microbiome in Brain Gliomas: A Systematic Review of Recent Evidence"

_ijms, 2025, doi:10.3390/ijms27010444_

Round 1

Reviewer 1 Report

Comments and Suggestions for Authors

I appreciate the opportunity to review the manuscript:  I appreciate the opportunity to review the manuscript:  ijms-4018287 titled: „ Emerging insights into the role of the microbiome in brain gliomas: a review of recent evidence”. Authors: Piotr Dubiński et.al.

Recommendation: Minor revision

General comments:

This review synthesizes findings from peerreviewed studies published in the last five years, including human cohorts, murine models, and Mendelian randomization analyses, selected using strict criteria focused on defined microbial taxa and signalling mechanisms.

In my opinion, the manuscript is interesting because although significant strides have been made in the molecular characterization of gliomas the understanding of glioma pathogenesis remains incomplete, regarding particularly environmental and systemic contributors. Moreover , a growing body of research underscores the role of the gut microbiota in shaping neurological health and disease. This is a very promising approach that may have clinical applications.

However, the manuscript submitted for review has certain limitations: it only discusses the results of a small number of scientific publications. Although these insights are promising, current evidence is constrained by small cohorts, limited model translatability, and methodological heterogeneity. Standardized, humanized models and integrated approaches are required to elucidate causal mechanisms and support the development of microbiome-targeted therapeutic strategies in glioma.

The manuscript is very well prepared, but I believe it lacks graphical diagrams that would facilitate understanding of individual subsections and enrich the manuscript. And my opinion manuscript requires these corrections.

Author Response

1st REVIEW REPORT

Dear Reviewer #1,

First of all, we would like to thank you for taking the time to read our article. Due to the comments, you sent us, we have tried to make corrections as best as possible. We hope that the changes introduced have helped us improve our work. We encourage you to read the entire content of the article, as we have made numerous corrections according to your recommendations. Thank you very much for your opinion and we look forward to any comments you may have.

Point 1: This review synthesizes findings from peer reviewed studies published in the last five years, including human cohorts, murine models, and Mendelian randomization analyses, selected using strict criteria focused on defined microbial taxa and signalling mechanisms. In my opinion, the manuscript is interesting because although significant strides have been made in the molecular characterization of gliomas the understanding of glioma pathogenesis remains incomplete, regarding particularly environmental and systemic contributors. Moreover, a growing body of research underscores the role of the gut microbiota in shaping neurological health and disease. This is a very promising approach that may have clinical applications.

Response regarding point 1: Thank you for your time and effort in reviewing our manuscript. We appreciate your comments and will incorporate them into the revision process without delay. Given the poor prognosis of glioma patients despite all currently available treatments, as well as the incomplete understanding of the disease’s etiopathogenesis and risk factors, expanding research to include the role of the microbiome in glioma genesis is scientifically warranted. This research direction remains innovative, as no clinically implemented diagnostic tests or therapeutic strategies based on microbiota modulation currently exist, even though emerging data suggest that such approaches may hold substantial future potential. Ensuring the reliability and highest quality of the publication is therefore essential, and we will achieve this by implementing all points raised in your review.

Point 2: However, the manuscript submitted for review has certain limitations: it only discusses the results of a small number of scientific publications. Although these insights are promising, current evidence is constrained by small cohorts, limited model translatability, and methodological heterogeneity. Standardized, humanized models and integrated approaches are required to elucidate causal mechanisms and support the development of microbiome-targeted therapeutic strategies in glioma.

Response regarding point 2: We agree with your assessment, it should be noted that current evidence in this field is indeed constrained by small cohorts, the limited translatability of existing experimental models and methodological heterogeneity. These factors restrict the strength of causal inference and the development of microbiota-oriented practical implications. We have revised the manuscript accordingly, incorporating the received suggestions to improve the presented content, with all additions marked using yellow-highlighted text.

Point 3: The manuscript is very well prepared, but I believe it lacks graphical diagrams that would facilitate understanding of individual subsections and enrich the manuscript. And my opinion manuscript requires these corrections.

Response regarding point 3: Acknowledging that visual representation enhances the comprehensibility of the material, we have adopted the received advice and introduced a set of tabular summaries. The summaries are presented in chapter 3 and 4 of the revised manuscript and are highlighted in yellow font. The manuscript is now supplemented by the following figures and tables:

Figure 1. PRISMA 2020 flowchart illustrating the selection process of studies included in this systemic review.

Tab. 1. Summary of microbial taxa identified in glioma patients and their associated role in glioma pathogenesis.

Taxonomy

Change in numerosity

Related effect

Research methods and source

Firmicutes

Tumour growth

Murine model (mainly) and cross-species projects, stool examination [10]

Bacteroidetes

Verrucomicrobia

Akkermansia muciniphila

Fusobacterium,
Fusobacterium nucleatum

Enhanced tumour progression, increased expression of proinflammatory chemokines CXCL1, CXCL2 and CCL2

16S rRNA sequencing combined with transcriptomics, metabolomics, IHC, multicolour immunofluorescence, and FISH; animal models [13]

Longibaculum

Pasteurella

Intestinimonas

Arthrobacter

Limosilactobacillus

Alcaligenaceae

Impaired maturation of cytotoxic natural killer (NK) cells infiltrating brain tissue, reduced mature CD27⁺/CD11b⁺ NK cell subset

Murine model, exposition to antibiotics [17]

Burkholderiaceae

Prevotellaceae

Rikenellaceae

Helicobacteraceae

Lachnospiraceae

Mutation in IDH1‑mutant gliomas, typically corresponding to LGG, rare in IDH1–wild‑type gliomas, uncommon in grade IV tumours

Oral microbiota examination
(saliva samples 16S rRNA sequencing analysis), stratified by malignancy grade,
low-grade gliomas
(LGG, WHO grades I-II)
and high-grade gliomas
(HGG, WHO grades III-IV),
presence of IDH1 mutation [21]

Enterobacteriaceae

Bacteroides

Prevotella

Capnocytophaga

Higher glioma grade

Leptotrichia

Porphyromonas

Haemophilus

Capnocytophaga

TM7x

Bergeyella

Presence of isocitrate dehydrogenase 1 (IDH1)

Capnocytophaga

Firmicutes

Adlercreutzia

Increased glioma risk

Human Mendelian randomization studies,
16S rRNA gene sequencing and host genotyping [22, 23, 24]

Prevotella

Catenibacterium

Coprobacter

Olsenella

Peptostreptococcaceae

Verrucomicrobia

Euryarchaeota

Anaerostipes

Faecalibacterium

Lachnospiraceae UCG004

Phascolarctobacterium

Streptococcus

Ruminococcaceae

Tab. 2. The GBA’s key microbial metabolites and neurotransmitters and their role in the glioma pathogenesis.

Molecule

Change in concentration

Biological role

Impact on glioma progression

Acetate

Crossing the BBB, decrease of proinflammatory Th1 and Th17 cells and increase of anti-inflammatory Tregs, neutrophil chemotaxis, enhancement of IL-10 secretion, inhibition of NF-κB signalling, suppression of proinflammatory cytokine production by myeloid cells

Propionate

Butyrate

GABA

Reduced proliferation and maintenance of cellular quiescence

Serotonin

Promotion of cell proliferation and migration, induction if differentiation, increase in the release of GDNF

Dopamine

Regulation of cell survival and proliferation, enhancement of the growth of spheroids enriched in cancer stem cells

Norepinephrine

Modulation of proliferation and inhibition of migration and invasion, maintenance of BBB integration

Acetylcholine

Tumour growth, angiogenesis, glioma stemness, tumour proliferation

Glutamate

Cell growth, enhancement of proliferation and migration, promotion of perivascular invasion, overall stimulation of tumour growth and invasiveness

Glycine

Promotion of angiogenesis and cellular stemness, a mediator contributing to glioma progression (in antibiotic-treated mice)

Tryptophan

Gliomas exploit the kynurenine pathway to suppress anti-tumour immune responses via upregulation of IDO1 and TDO, modulated by microbial signals

Reviewer 2 Report

Comments and Suggestions for Authors

In this review article, the authors discuss the emerging role of the microbiome in the progression and treatment of brain gliomas, particularly glioblastoma multiforme, highlighting recent research findings and the need for further studies.

The review is interesting and well organized; however, several aspects can be improved.

Major Comments

  1. It would be relevant to indicate the search strategy and the combination of keywords and MeSH terms used to identify the 17 works.
  2. A figure summarizing the general conceptual overviews and the specific mechanistic findings, both preclinical and clinical, would be helpful.
  3. A visual representation comparing healthy individuals and glioma patients, highlighting the key microbial taxa and their relative abundance, would be very useful. This should include a depiction of the changes in microbial composition associated with glioma.
  4. In addition, a diagram illustrating the bidirectional communication between the gut microbiota and the central nervous system (CNS) would be valuable. It can include key pathways such as immune modulation (e.g., Tregs, cytokines), metabolic signaling (e.g., SCFAs, neurotransmitters), and neuroendocrine interactions (e.g., HPA axis). The figure can also highlight how dysbiosis impacts glioma progression through immune suppression, neuroinflammation, and blood–brain barrier dysfunction.
  5. A table summarizing the microbial taxa identified in glioma patients and their associated roles in glioma pathogenesis would also be useful. Columns: Microbial Taxa (Phylum, Genus, Species); Change in Abundance (Increased/Decreased); Associated Effects (e.g., immune suppression, neuroinflammation, angiogenesis, protective effects); Supporting Studies.
  6. Another table could include microbial metabolites and their roles in glioma pathogenesis, outlining the key metabolites produced by the microbiota and their impact on glioma progression. Columns: Metabolite (e.g., SCFAs, GABA, serotonin, dopamine); Change in Levels (Increased/Decreased); Biological Role (e.g., immune modulation, blood–brain barrier integrity, angiogenesis); Impact on Glioma (e.g., promotes progression, inhibits progression).

Author Response

1st REVIEW REPORT

Dear Reviewer #2,

Thank you very much for dedicating your valuable time to review our manuscript and for providing insightful feedback. We sincerely appreciate your constructive comments, which we consider highly relevant and important for a proper understanding of our research. We have undertaken a comprehensive revision process to address the points raised. We confirm that we have analysed your comments in detail and have implemented corrections according to your recommendations. We encourage a review of the entire revised manuscript to fully observe how the implemented corrections have collectively enhanced the quality and clarity of the content. We look forward to any further observations you may have.

Point 1: It would be relevant to indicate the search strategy and the combination of keywords and MeSH terms used to identify the 17 works.

Response regarding point 1: We acknowledge the validity of your observation regarding the necessity of transparent reporting on the methodology. Acknowledging that the inclusion of the search strategy and the combination of keywords and MeSH terms enhances the reproducibility and scholarly rigour of the work, we provide the literature selection strategy below (marked in yellow in the article in chapter 3):

A systematic literature search for publications regarding microbiome and brain gliomas was conducted, spanning the period from 2020 to 2025. The search was conducted in medical literature, analysis, and retrieval system on-line (MEDLINE) and the Cochrane Central Register of Controlled Trial (Cochrane) databases. The following search phrases were used: MEDLINE (via PubMed) - (‘microbiota’ [MeSH] or ‘microbiome’) and (‘glioma’ [MeSH] or ‘glioblastoma’ [MeSH]) yielding 56 records; and Cochrane - (‘microbiome’ or ‘microbiota’ and ‘gliomas’), yielding 0 records. The inclusion criteria comprised high-impact reviews, fundamental in vitro and in vivo experimental studies, including animal models research, statistical analyses based on Mendelian Randomization, and clinical cohort studies. The research covers phenomena related to dysbiosis in the course of gliomas, primarily microbial taxonomy, metabolic pathways, metabolomic alterations, and immunomodulation. The exclusion criteria constituted articles describing patient populations other than those with gliomas and reports that mainly dealt with aspects related to the surgical technique. The search strategy followed PRISMA 2020 guidelines and is illustrated in Figure 1.

Figure 1. PRISMA 2020 flowchart illustrating the selection process of studies included in this systemic review.

Point 2:  A figure summarizing the general conceptual overviews and the specific mechanistic findings, both preclinical and clinical, would be helpful.

Point 3: A visual representation comparing healthy individuals and glioma patients, highlighting the key microbial taxa and their relative abundance, would be very useful. This should include a depiction of the changes in microbial composition associated with glioma.

Point 4: In addition, a diagram illustrating the bidirectional communication between the gut microbiota and the central nervous system (CNS) would be valuable. It can include key pathways such as immune modulation (e.g., Tregs, cytokines), metabolic signalling (e.g., SCFAs, neurotransmitters), and neuroendocrine interactions (e.g., HPA axis). The figure can also highlight how dysbiosis impacts glioma progression through immune suppression, neuroinflammation, and blood–brain barrier dysfunction.

Response regarding points 2-4:  We are sincerely grateful for your insightful suggestion and fully concur that visual representation significantly aids in the consolidation of complex scientific content. We confirm that, upon expanding our team to include personnel proficient in handling advanced visual materials, we will prioritize the development and integration of this comprehensive graphical element to substantially enrich the manuscript. Thank you for this constructive guidance for the future development of our work. The graphic changes were introduced later in the work and marked in yellow in the article.

Point 5: A table summarizing the microbial taxa identified in glioma patients and their associated roles in glioma pathogenesis would also be useful. Columns: Microbial Taxa (Phylum, Genus, Species); Change in Abundance (Increased/Decreased); Associated Effects (e.g., immune suppression, neuroinflammation, angiogenesis, protective effects); Supporting Studies.

Response regarding point 5: We concur with your suggestion that a dedicated summary table would significantly enhance the accessibility and organization of the reported data. Acknowledging the clear benefit of structuring this specific information, we have incorporated the table below presenting the summary of microbial taxa identified in glioma patients and their associated role in glioma pathogenesis. The changes were introduced in the work (Chapter 4) and marked in yellow in the article.

Tab. 1. Summary of microbial taxa identified in glioma patients and their associated role in glioma pathogenesis

Taxonomy

Change in numerosity

Related effect

Research methods and source

Firmicutes

Tumour growth

Murine model (mainly) and cross-species projects, stool examination [10]

Bacteroidetes

Verrucomicrobia

Akkermansia muciniphila

Fusobacterium,
Fusobacterium nucleatum

Enhanced tumour progression, increased expression of proinflammatory chemokines CXCL1, CXCL2 and CCL2

16S rRNA sequencing combined with transcriptomics, metabolomics, IHC, multicolour immunofluorescence, and FISH; animal models [13]

Longibaculum

Pasteurella

Intestinimonas

Arthrobacter

Limosilactobacillus

Alcaligenaceae

Impaired maturation of cytotoxic natural killer (NK) cells infiltrating brain tissue, reduced mature CD27⁺/CD11b⁺ NK cell subset

Murine model, exposition to antibiotics [17]

Burkholderiaceae

Prevotellaceae

Rikenellaceae

Helicobacteraceae

Lachnospiraceae

Mutation in IDH1‑mutant gliomas, typically corresponding to LGG, rare in IDH1–wild‑type gliomas, uncommon in grade IV tumours

Oral microbiota examination
(saliva samples 16S rRNA sequencing analysis), stratified by malignancy grade,
low-grade gliomas
(LGG, WHO grades I-II)
and high-grade gliomas
(HGG, WHO grades III-IV),
presence of IDH1 mutation [21]

Enterobacteriaceae

Bacteroides

Prevotella

Capnocytophaga

Higher glioma grade

Leptotrichia

Porphyromonas

Haemophilus

Capnocytophaga

TM7x

Bergeyella

Presence of isocitrate dehydrogenase 1 (IDH1)

Capnocytophaga

Firmicutes

Adlercreutzia

Increased glioma risk

Human Mendelian randomization studies,
16S rRNA gene sequencing and host genotyping
[22, 23, 24]

Prevotella

Catenibacterium

Coprobacter

Olsenella

Peptostreptococcaceae

Verrucomicrobia

Euryarchaeota

Anaerostipes

Faecalibacterium

Lachnospiraceae UCG004

Phascolarctobacterium

Streptococcus

Ruminococcaceae

Point 6: Another table could include microbial metabolites and their roles in glioma pathogenesis, outlining the key metabolites produced by the microbiota and their impact on glioma progression. Columns: Metabolite (e.g., SCFAs, GABA, serotonin, dopamine); Change in Levels (Increased/Decreased); Biological Role (e.g., immune modulation, blood–brain barrier integrity, angiogenesis); Impact on Glioma (e.g., promotes progression, inhibits progression).

Response regarding point 6: We fully agree that a complementary table summarizing the identified microbial metabolites and their distinct roles in glioma pathogenesis is highly valuable. Consequently, we have incorporated a dedicated table below presenting the relevant data. The changes were introduced in the work (Chapter 4) and marked in yellow in the article.

Tab. 2. The GBA’s key microbial metabolites and neurotransmitters and their role in the glioma pathogenesis

Molecule

Change in concentration

Biological role

Impact on glioma progression

Acetate

Crossing the BBB, decrease of proinflammatory Th1 and Th17 cells and increase of anti-inflammatory Tregs, neutrophil chemotaxis, enhancement of IL-10 secretion, inhibition of NF-κB signalling, suppression of proinflammatory cytokine production by myeloid cells

Propionate

Butyrate

GABA

Reduced proliferation and maintenance of cellular quiescence

Serotonin

Promotion of cell proliferation and migration, induction if differentiation, increase in the release of GDNF

Dopamine

Regulation of cell survival and proliferation, enhancement of the growth of spheroids enriched in cancer stem cells

Norepinephrine

Modulation of proliferation and inhibition of migration and invasion, maintenance of BBB integration

Acetylcholine

Tumour growth, angiogenesis, glioma stemness, tumour proliferation

Glutamate

Cell growth, enhancement of proliferation and migration, promotion of perivascular invasion, overall stimulation of tumour growth and invasiveness

Glycine

Promotion of angiogenesis and cellular stemness, a mediator contributing to glioma progression (in antibiotic-treated mice)

Tryptophan

Gliomas exploit the kynurenine pathway to suppress anti-tumour immune responses via upregulation of IDO1 and TDO, modulated by microbial signals